# The Ionome–Hormone–Flavonoid Network Shapes Genotype-Dependent Yield Adaptation in Sugarcane

**DOI:** 10.3390/plants14203181

**Published:** 2025-10-16

**Authors:** Qinyu Lu, Shimiao Chen, Bin Shan, Ailin Wei, Yuhuan Luo, Lanfang Wu, Qiang Jiang, Zhendong Chen

**Affiliations:** 1Guangxi Baise Modern Agriculture Technology Research and Extension Center, Management Committee of Baise National Agricultural Science and Technology Zone of Guangxi, Baise 533612, China; qiang881028@126.com; 2Guangxi Subtropical Crops Research Institute, Nanning 530001, China; chenshimiao@gxaas.net (S.C.); zzbj219@163.com (B.S.); 3Baise Institute of Agricultural Sciences, Baise 533612, China; bstsny@163.com (A.W.); bsnyy@163.com (L.W.); 4Seed Station of the Bureau of Agriculture and Rural Affairs, Tianyang District, Baise 533612, China; lyhuan1688@163.com; 5Guangxi Academy of Agricultural Science, Nanning 530001, China

**Keywords:** nutrient homeostasis, phytohormone, flavonoids, constitutive defense posture, multivariate analysis

## Abstract

Sugarcane productivity varies widely among genotypes, but the biochemical traits underlying these differences remain poorly characterized. In this study, six contrasting sugarcane cultivars were profiled to investigate how ionomic, hormonal, flavonoid, and photosynthetic pigment signatures are associated with yield and sucrose accumulation. Morphological traits and field performance revealed marked genotypic variation, with ZZ14 and GL1215 achieving the highest yields and sugar content, while GT59 and GT60 performed less favorably. Multivariate analyses of ionomic data showed that potassium, magnesium, and calcium were consistently enriched in high-yield cultivars, whereas sodium, boron, and manganese were negatively associated with growth traits. Hormone profiling revealed that high-yielding genotypes utilize diverse strategies: while the high-yielding GL1215 achieved superior sugar content with the lowest levels of growth-promoting hormones, the LT1790 genotype, despite having the highest levels of these hormones, showed suboptimal yield due to a costly trade-off with its hyperactive defense system. Flavonoid analysis indicated that LT1790 contained the highest levels of Quercetin, rutin, and caffeic acid, suggesting enhanced antioxidant capacity, whereas GT59 preferentially accumulated chlorogenic acid. Canonical correlation analysis confirmed that nutrient balance and metabolite composition strongly correlated with plant height, stem diameter, and sugar concentration. Together, these results suggest that high-yield sugarcane genotypes achieve a superior metabolic balance, combining efficient nutrient uptake and robust antioxidant capacity with a favorable hormone profile that promotes strong growth without triggering a costly constitutive defense system.

## 1. Introduction

Sugarcane is a key industrial crop predominantly cultivated in tropical and subtropical regions, with Brazil, India, and China collectively accounting for over 60% of global production [1]. It supplies approximately 80% of the world’s sucrose and contributes 40% of Brazil’s automotive fuel through ethanol production [2]. Beyond its role in food and energy, sugarcane is a renewable resource for bioplastics, pharmaceuticals, and construction materials [3]. Over the past two decades, global sugarcane yields have shown gradual improvement, attributed mainly to genetic advances and the adoption of precision agriculture technologies [4].

Genetic potential forms the foundation of sugarcane yield and biomass. Owing to its highly polyploid genome, sugarcane exhibits broad genetic diversity that can be exploited for yield improvement. Statistical analyses using Restricted Maximum Likelihood (REML) and Best Linear Unbiased Prediction (BLUP) have reported very high heritability (>85%) for traits related to commercial cane sugar, with a genetic coefficient of variation (GCV) of 16.6–21.6%, whereas germination traits show much lower heritability (~35%) [5]. Quantitative trait analyses (QTAs) for stalk diameter and internode length further support the use of genomic selection strategies such as t-BLUP (e.g., cultivar CP 00-1101) [6].

Beyond genetics, soil conditions and nutrient management also play critical roles in determining sugarcane productivity. For example, one study found that the application of 120 kg K_2_O/ha polyhalite significantly enhanced ratoon sprouting by 34.9%, reduced pest incidence by 28.6%, and, when combined with optimized nitrogen fertilization, increased stalk yield by 4.9% [7]. These results emphasize the importance of integrating balanced nutrient regimes into breeding and cultivation programs.

Sugarcane is sensitive to abiotic stress, and its physiological responses can greatly influence yield stability. Under drought or flooding, photosystem II (PSII) efficiency typically declines; however, the genotype Khon Kaen 3 (KK3) maintained higher stem weight, and potassium supplementation helped preserve chlorophyll as well as the activities of NADP-dependent malate dehydrogenase (NADP-MDH) and Phosphoenolpyruvate carboxylase (PEPC) [8]. Agronomic practices such as fenlong-ridging improved soil porosity by 15%, enhanced Ribulose-1,5-bisphosphate carboxylase (RuBPC) and PEPC activities by 20–25%, and increased plant-cane yield by about 12% [9].

Pest management and climate adaptability also shape long-term yield outcomes. Integrated pest management combined with Bru1 resistance reduced borer damage by 23–28% and allowed yields to reach 89.43 t/ha [10]. Ratooning ability also differs across varieties, with N25 showing a reduction of 2.74 t/ha per ratoon cycle compared to only 1.38 t/ha in N36 [11]. Climate models under Representative Concentration Pathway (RCP) 4.5, a scenario where global greenhouse gas emissions peak around 2040 and then decline, predict sucrose losses. Genotype–environment interaction analyses further indicate that thermal time and solar radiation account for 40–60% of yield variation, highlighting stable genotypes such as N36 as promising candidates for future climate conditions [12]. Together, these findings illustrate the critical interplay of genetic, environmental, and management factors in sustaining sugarcane productivity.

Ionomic profiling has revealed intricate elemental interactions and hormone-mediated signaling networks that regulate plant growth, stress responses, and environmental adaptation. In Arabidopsis, sulfur–phosphate cross-talk coordinates foliar sulfur content, sulfate uptake, and phosphate homeostasis with downstream metabolic pathways such as glucosinolate and glutathione biosynthesis, emphasizing sulfur’s dual role in nutrition and defense [13]. Under salt stress, sustaining a high K^+^/Na^+^ ratio is vital for osmotic balance and enzyme stability, with High-affinity K^+^ Transporter (*HKT*) and Salt Overly Sensitive (*SOS*) transporters facilitating Na^+^ sequestration while preserving K^+^ levels [14]. In wheat, optimal Ca^2+^/Na^+^ and Mg^2+^/Na^+^ ratios, maintained through electrochemical gradients generated by H-ATPases, enhance salt tolerance by selectively permitting beneficial cation uptake while excluding toxic Na^+^ [15].

Heavy metals such as Pb^2+^ and Cu^2+^ can disrupt this ionic equilibrium by displacing essential nutrients at transport and enzyme binding sites, impairing seed germination and early seedling growth [16]. Hormonal regulation plays a pivotal role in maintaining ionic homeostasis: abscisic acid (ABA), auxin (IAA), salicylic acid, and melatonin modulate transporter expression and proton pump activity to coordinate the compartmentalization of toxic ions and the uptake of essential nutrients [15]. Auxin gradients, via PIN-mediated transport, influence root Ca^2+^ signaling and the polarization of H-ATPases, enhancing K^+^ and NO_3_^−^ uptake under nutrient-deficient conditions [17]. ABA-induced Ca^2+^ transients, in turn, regulate stomatal closure through Slow Anion Channel-Associated 1 (*SLAC1*)-dependent K^+^ channel activation [18]. Cytokinins adjust root-to-shoot ion distribution via HAK K^+^ transporters [19], while ethylene promotes Fe acquisition through IRT1 upregulation under iron deficiency [20]. These findings suggest that phytohormones and flavonoids are central regulatory factors orchestrating plant ionome-related cross-talk.

Plant hormones orchestrate sugarcane growth, development, and stress adaptation through interconnected signaling networks that regulate biomass accumulation, structural development, and environmental resilience. Auxins (indole-3-acetic acid [IAA] and indole-3-butyric acid [IBA]) are pivotal for cell differentiation and biomass yield: one study found that high-biomass genotypes exhibited elevated IAA levels, and exogenous IBA application under drought increased relative water content and plant height [21]. Notably, sugarcane vinasses contain micromolar concentrations of bioactive IAA and ABA, implying that fertigation practices may confer hormonal benefits beyond nutrient supply [22].

Gibberellins (GA_3_) drive internode elongation and stem growth; one study reported that GA_3_ treatment, particularly when combined with melatonin, enhanced antioxidant enzyme activities (SOD, POD), reduced reactive oxygen species, and mitigated drought-induced growth inhibition [23]. Abscisic acid (ABA) functions as a master stress hormone: a study showed that pre-treatment with 100 µM ABA elevated endogenous ABA by ~42%, boosted proline by 45–80% over 6–9 days, and preserved cellular integrity under water deficit [24]. A study showed that Brassinosteroids (brassinolide at 5 mg/L) significantly improved shoot proliferation, leaf number, and fresh weight in vitro and synergized with auxin and cytokinin to promote tracheary element differentiation for vascular development [25].

In mediating biotic and abiotic stress responses, one study reported that 1 mM SA reduced Sorghum mosaic virus populations in cultivars ROC22 and Xuezhe, modulating 2999 genes enriched in phenylpropanoid biosynthesis and pathogen-interaction pathways, and enhanced early-stage salinity tolerance through osmotic adjustment and antioxidant defense [26]. During internode elongation, transcriptomic analyses revealed an enrichment of “zeatin biosynthesis,” “nitrogen metabolism,” and “plant hormone signal transduction” pathways, indicating coordinated roles for cytokinins alongside other hormones in transitioning from cell wall synthesis to sucrose storage [27].

An integrative co-expression network analysis highlighted key transcription factor families (AP2/ERF, Dof) and repressor genes (Aux/IAA, PP2C, JAZ) that translate hormonal signals into growth and stress-response pathways, with high-biomass genotypes showing elevated IAA and ABA but reduced jasmonic acid levels [21]. Exogenous applications of hormone combinations (melatonin + GA_3_; melatonin + IBA + GA_3_) delivered comprehensive protection against drought by enhancing chlorophyll retention, antioxidant capacity, and osmolyte accumulation [23].

Flavonoids in sugarcane function as versatile secondary metabolites that integrate growth regulation, stress defense, and environmental adaptation. Their potent antioxidant activity mitigates oxidative damage under abiotic stresses such as UV-B radiation, safeguarding membrane integrity and supporting biomass accumulation [28]. In stalk rind tissues, a study found that cultivar-specific flavonoid profiles correlated with higher free-radical-scavenging capacity and enhanced tolerance to drought and high-light conditions [29].

Beyond direct antioxidation, flavonoids contribute to pathogen defense. It was found that elevated flavonoid levels in resistant cultivars inhibited fungal and bacterial colonization, and key flavonoid biosynthetic genes were up-regulated upon infection, underscoring their role in inducible defense responses [30]. Genetic engineering of flavonoid pathway enzymes (e.g., F3H) has further demonstrated increased disease resistance in transgenic plants [31].

Flavonoids also mediate plant–microbe cross-talk in the rhizosphere and endosphere. Exuded flavonoids recruit and modulate beneficial bacteria, producing hydrolytic enzymes and antifungal metabolites that enhance nutrient availability and suppress pathogens [32]. These mutualistic interactions improve phosphorus solubilization and nitrogen fixation, contributing to overall plant vigor.

Hormonal regulation tightly controls flavonoid biosynthesis and signaling. Studies have shown that auxin treatments elevated flavonoid accumulation and promoted root cell wall remodeling during adventitious rooting, while cytokinins and jasmonates coordinated the expression of flavonoid pathway genes under developmental and stress cues [31]. Meanwhile, flavonoids can influence hormone transport and perception, creating feedback loops that fine-tune growth and defense.

At the post-transcriptional level, miRNAs target structural and regulatory genes in the flavonoid network, modulating metabolite flux in response to developmental stage and environmental stimuli [33]. Under heavy-metal exposure and elevated temperatures, shifts in flavonoid composition enable metal chelation and maintenance of redox balance, facilitating acclimation to soil contamination and heat stress [34].

Flavonoids also affect crop quality traits. In sugarcane and other fruiting crops, dynamic changes in flavonoid content during maturation determine antioxidant potential, color, and taste, directly influencing market value [35].

Emerging evidence suggests that plant hormones, flavonoids, and the ionome constitute an integrated regulatory network that governs growth, development, constitutive defense posture, and crop quality. Although individual binary interactions, such as auxin–flavonoid cross-talk [36,37] and Ca^2+^–ABA coordination in stomatal control [38], have been elucidated, a holistic, systems-level understanding is still missing. Future studies should combine high-throughput ionomic profiling, targeted hormone quantification, flavonoid metabolomics, and network modeling to capture synergistic and antagonistic interactions across these layers. Insights from such integrative approaches will enable precision breeding and tailored exogenous treatments to optimize yield, nutritional value, and stress tolerance in sugarcane and other crops [39]. As climate change and food security challenges intensify, deciphering the hormone–flavonoid–ionome nexus will be crucial for designing sustainable strategies that harness plant adaptive mechanisms to enhance agricultural productivity.

To investigate the biochemical underpinnings of these differences, six sugarcane cultivars with contrasting yield and sucrose content were profiled. The elemental, hormonal, and flavonoid profiles for each genotype were quantified, and their corresponding field-level cane yield and juice Brix levels were measured. This integrated approach allowed for a direct assessment of how these biochemical signatures correlate with agronomic performance, providing critical insights into the key factors that differentiate high-yield, high-sugar cultivars from their lower-performing counterparts.

## 2. Results

### 2.1. Genotype-Dependent Variation in Sucrose Content and Yield Among Different Varieties

The sugarcane yield, sugar content, and key morphological traits of different varieties are summarized in Figure 1. Plant height varied significantly across the six genotypes, ranging from 294.39 ± 4.41 cm in GT59 to 337.30 ± 3.89 cm in ZZ14. Intermediate heights were observed in GL05136 (311.90 ± 2.71 cm), GL1215 (332.39 ± 2.94 cm), and LT1790 (324.15 ± 3.57 cm), all of which were significantly taller than the two shortest genotypes, GT59 and GT60. In contrast, stem diameter exhibited only minor variation, from 24.29 ± 0.22 mm in GT60 to 25.47 ± 0.52 mm in GT59, with no significant differences detected among the varieties.

Sugar content (Brix) ranged from 16.12 ± 0.37% in GT59 to 18.73 ± 0.28% in GL1215, with intermediate levels in GL05136 (16.46 ± 0.25%), GT60 (17.00 ± 0.27%), ZZ14 (17.08 ± 0.17%), and LT1790 (17.55 ± 0.22%). Notably, GL1215 had the highest sugar concentration, suggesting a potential advantage for high-sugar yield cultivation.

Yield per hectare exhibited the most significant variability, from 90.99 ± 1.84 t in GT59 to 168.92 ± 2.51 t in ZZ14. GL1215 (146.47 ± 0.97 t), GL05136 (124.95 ± 1.23 t), and LT1790 (94.26 ± 2.18 t) also significantly outperformed the lower-yielding GT60 (113.90 ± 0.76 t). These results suggest that the substantial differences among genotypes in yield, sugar content, and key morphological traits make varietal selection a critical factor for improving sugarcane productivity and quality.

### 2.2. Ionomic Profiling and Multivariate Analysis of Sugarcane Genotypes

Principal component analysis (PCA) of ionomic profiles across six sugarcane varieties (Figure 2a) revealed that the first two principal components (PC1 and PC2) together accounted for 80.3% of the total variance, with PC1 explaining 61.5% and PC2 capturing 18.8%. Along PC1, the genotypes GT59, GT60, and GL05136 clustered positively, reflecting higher overall mineral content, while GL1215 and ZZ14 were positioned on the negative side, indicating distinct elemental accumulation patterns. LT1790 occupied an intermediate position near the origin, suggesting a more balanced ionomic profile. GT59 showed particularly tight clustering among its biological replicates, indicating greater phenotypic stability than the more variable profiles observed in GL1215 and ZZ14.

To further resolve genotype-specific elemental signatures, orthogonal partial least squares discriminant analysis (OPLS-DA) models were constructed for five pairwise genotype comparisons: GT59 vs. LT1790, GL05136 vs. GL1215, GL05136 vs. GT60, GL05136 vs. ZZ14, and GT59 vs. GT60. All models exhibited robust predictive power (Q^2^cum ≥ 0.5) without signs of overfitting, confirming their reliability for discriminating between genotypes. VIP (variable importance in projection) scores highlighted several key elements consistently driving differentiation across these comparisons, including potassium (K), magnesium (Mg), calcium (Ca), manganese (Mn), and barium (Ba). For example, in the GT59 vs. LT1790 comparison, the top discriminators included K (VIP = 3.734), Mg (2.981), Ca (2.797), and Mn (2.480). Similarly, the GL05136 vs. GL1215 model prioritized K (766.4 nm, 4.547), K (769.8 nm, 3.996), Mg (3.037), Ca (2.117), and Ba (1.887), while the GL05136 vs. GT60 model ranked Mg (5.818), Ba (2.734), Ca (2.214), Mn (1.645), and Na (1.592) as the most influential features. Consistent prioritization of these elements across multiple genotype comparisons underscores their central roles in defining sugarcane ionomic differentiation, reflecting both genotypic nutrient uptake efficiency and potential stress adaptation mechanisms.

Importantly, PCA score plots for genotype pairs where OPLS-DA models exhibited Q^2^cum < 0.5 showed near-complete overlap of replicates along PC1 and PC2, indicating minimal ionomic variation within these groups. This supports the decision to omit these overfitted comparisons, as their exclusion did not significantly reduce the interpretative power of the overall analysis.

Leaf chlorophyll content analysis across six sugarcane genotypes revealed significant genotype-dependent variation (Figure 3). Chlorophyll A content ranged from a minimum of 1.4449 ± 0.0218 mg/g in GL1215 to a maximum of 3.4259 ± 0.0305 mg/g in GT60, with LT1790 also showing elevated levels (3.2999 ± 0.0683 mg/g), significantly higher than the other genotypes. Chlorophyll B followed a similar trend, with GT60 (1.6853 ± 0.0256 mg/g) and LT1790 (1.5490 ± 0.0231 mg/g) displaying the highest concentrations, while GL1215 exhibited the lowest (0.6497 ± 0.0076 mg/g). Consequently, total chlorophyll content peaked in LT1790 (5.0791 ± 0.0767 mg/g) and GT60 (5.0203 ± 0.0773 mg/g), significantly surpassing the other genotypes, while GL1215 had the lowest total chlorophyll (2.0651 ± 0.0253 mg/g), as determined from absorbance at 645 and 663 nm rather than by summing chlorophyll a and b.

Carotenoid levels also varied substantially, ranging from 48,139.5 ± 484.1 µg/g in GL1215 to 82,572.8 ± 1869.4 µg/g in LT1790, indicating a broader capacity for photoprotection and antioxidant function in the latter. The significant pairwise differences (*p* < 0.05) suggest that GT60 and LT1790 possess enhanced pigment accumulation, which may contribute to superior photosynthetic efficiency relative to other genotypes.

### 2.3. Phytohormone Profiling of Different Genotypes Reveals Genotype-Dependent Variations

Endogenous hormone profiling of six sugarcane genotypes revealed significant genotype-dependent differences in phytohormone content (Figure 4). Abscisic acid (ABA) levels varied substantially, ranging from a low of 1325.15 ± 28.39 ng/g in GL1215 to a high of 2289.18 ± 32.33 ng/g in LT1790, reflecting potential differences in stress tolerance and dormancy regulation. 1-Aminocyclopropane-1-carboxylic acid (ACC), a precursor of ethylene, also showed marked variation, with LT1790 exhibiting the lowest content (102,695.04 ± 1623.35 ng/g) and ZZ14 the highest (189,405.30 ± 2949.88 ng/g), possibly indicating differential ethylene biosynthetic activity.

Brassinosteroid (Br) concentrations peaked in ZZ14 (2988.85 ± 19.66 ng/g), suggesting enhanced cell elongation and growth signaling in this genotype. At the same time, GT60 had the lowest Br levels (1105.70 ± 31.14 ng/g), potentially reflecting reduced brassinosteroid biosynthesis or heightened catabolism. Gibberellin (GA_3_) levels also differed significantly, with LT1790 having the highest content (903.26 ± 6.04 ng/g), indicative of enhanced shoot elongation capacity, while GL1215 exhibited the lowest levels (566.94 ± 5.05 ng/g).

Indole-3-acetic acid (IAA), a key regulator of cell elongation and bud outgrowth, was most abundant in LT1790 (2025.54 ± 23.72 ng/g) and lowest in GL1215 (1276.41 ± 29.02 ng/g), suggesting a potential link between high IAA levels and vigorous tillering. Similarly, salicylic acid (SA), known for its role in systemic acquired resistance, mirrored ABA trends, ranging from 1192.15 ± 16.91 ng/g in GL1215 to 2010.73 ± 23.70 ng/g in LT1790.

Trans-zeatin (tZ), a cytokinin involved in cell division and shoot formation, showed a contrasting pattern, peaking in GT60 (2488.56 ± 22.37 ng/g) and reaching its lowest levels in GT59 (1583.78 ± 22.47 ng/g), suggesting differential cytokinin regulation among the genotypes.

Collectively, these hormone profiles—particularly the elevated ABA, GA_3_, IAA, and SA in LT1790 and high tZ in GT60—highlight the critical role of phytohormonal regulation in shaping the distinct physiological and developmental traits observed among these sugarcane genotypes.

### 2.4. Genotype-Dependent Variation in Flavonoid Profiles of Different Sugarcane Varieties

Flavonoid profiling revealed significant genotype-dependent differences (Figure 5). LT1790 consistently exhibited the highest levels of several key flavonoids, including Astragalin (1904.83 ± 22.50 ng/g), Caffeic acid (2225.52 ± 27.20 ng/g), Quercetin (2192.44 ± 39.54 ng/g), Quinic acid (294,597.33 ± 8721.40 ng/g), Rosmarinic acid (2130.14 ± 30.08 ng/g), and Rutin (2737.08 ± 37.49 ng/g), indicating a broad capacity for antioxidant and photoprotective responses.

In contrast, GT59 exhibited the highest Chlorogenic acid content (3,173,718.94 ± 38,818.23 ng/g), suggesting a genotype-specific preference for this phenolic compound, which is often associated with enhanced UV protection and defense against oxidative stress. Sinapic acid was predominantly accumulated in ZZ14 (1566.38 ± 23.33 ng/g) and GT59 (1555.29 ± 22.25 ng/g), further highlighting these genotypes’ unique secondary metabolite profiles.

Interestingly, Astragalin and Coumaric acid were undetectable in GL05136 and GT59, suggesting possible metabolic trade-offs or pathway suppression in these varieties. Meanwhile, GL1215 and GT60 generally exhibited intermediate levels across most flavonoids, reflecting a more balanced but less specialized flavonoid profile.

These distinct accumulation patterns suggest that LT1790—and, to a lesser extent, GT59—may possess enhanced antioxidant and stress-protective capacities, potentially conferring greater resilience under biotic and abiotic stress conditions compared with the other sugarcane cultivars.

### 2.5. CCA of Yield Components with Ionomic and Metabolite Data in Different Sugarcane

Canonical Correlation Analysis (CCA) was employed to visualize the integrated relationships between ionomic (group I: Figure 6a,b) and broader metabolite (group M: Figure 6c,d) profiles, which explain variation in key yield components. The analysis revealed a clear, overarching trend: higher levels of endogenous growth hormones are generally associated with superior agronomic performance across the studied genotypes. Notably, the genotype LT1790 emerged as a significant exception to this rule, as its positioning in the CCA plot visually demonstrates a profile overwhelmingly dominated by a constitutive defense signature, which counteracts its growth-promoting potential. This dominant, nearly unidimensional trend was particularly evident in the metabolite-based CCA, where the first axis (CCA1) accounted for 99.39% of the total explained variance, with the second axis (CCA2) capturing just 0.36%. In contrast, the ion-based CCA explained a still substantial but less extreme 75.7% on CCA1 and 0.21% on CCA2, indicating that while the ion data also supports this primary gradient, it captures additional modest residual variation.

In the ion CCA biplot, potassium (K) emerged as the principal driver along CCA1, exhibiting a long vector directed toward the positive axis, reflecting its strong positive correlation with taller plants, thicker stems, and higher sugar content. In contrast, boron (B), sodium (Na), manganese (Mn), and silicon (Si) had significant negative loadings on CCA1, indicating that genotypes enriched in these ions tend to have lower values for these same traits. Magnesium (Mg) and Na also contributed to CCA2, creating a minor secondary gradient that slightly separated GT60 from the main cluster, though this axis accounted for less than 1% of the overall variance and thus had a limited impact on the broader yield-trait pattern.

The metabolite CCA biplot mirrored this pattern with even greater intensity: growth-related compounds such as gibberellic acid (GA_3_), abscisic acid (ABA), salicylic acid (SA), astragalin, caffeic acid, and Quercetin tightly clustered along the positive direction of CCA1, each with high loadings (≥+5.5) and long vector lengths (~5.5–6.3 units). These metabolites were strongly associated with superior growth traits. In contrast, chlorogenic acid had a strong negative loading (≈−5.4), aligning with samples that exhibited lower growth performance. Similarly to the ion analysis, the second axis explained only a small fraction (0.36%) of the variance, lacking significant discriminatory power.

Sample ordination in both CCAs consistently revealed a continuous one-dimensional ranking of genotypes. In the ion CCA, GT59 and LT1790 occupied the positive extreme (high K, robust growth), while ZZ14 and GL1215 clustered at the negative extreme (high B, Na, Mn, Si, weaker growth traits). GL05136 occupied an intermediate position, while GT60 was slightly offset along CCA2, reflecting its unique ionic profile. In the metabolite CCA, samples similarly spanned a spectrum from high overall metabolite enrichment (positive CCA1) to higher chlorogenic acid and lower growth-promoting metabolites (negative CCA1), with GT60 again emerging as a mild outlier on CCA2.

In summary, both CCAs provide a dominant single-axis separation of sugarcane genotypes, driven primarily by a K-versus-other-ion gradient or a coordinated metabolite axis, respectively. These primary canonical axes were closely associated with variation in plant height, stem diameter, and sugar content.

## 3. Discussion

The present study demonstrates that sugarcane genotypes exhibit pronounced differences in ionomic profiles, hormone levels, flavonoid content, and photosynthetic pigments, and that these metabolic traits correlate with key agronomic performance indicators. Using canonical correlation analysis (CCA), it was found that multivariate metabolite/ionome signatures were strongly associated with yield-related traits (plant height, stem diameter, and sugar content). These results highlight a complex, genotype-dependent interplay between nutrition, hormonal regulation, and secondary metabolism driving sugarcane growth and productivity. Below, the findings of this study are compared with those of prior studies, and the implications are discussed for nutrient-use efficiency, hormone-regulated growth, flavonoid-mediated stress tolerance, and breeding high-yield, resilient cultivars.

### 3.1. Ionomic Composition and Nutrient-Use Efficiency

The significant variation observed in the accumulation of key mineral elements—including the macronutrients potassium (K), magnesium (Mg), and calcium (Ca)—among the genotypes reveals the diversity in their nutrient uptake efficiency and physiological adaptation strategies. The essential role of potassium in osmotic regulation and sucrose transport was strongly supported, as genotypes with higher K accumulation consistently exhibited superior stem growth and sugar content, aligning with previous findings that K supports photosynthesis, enzyme activation, and sugar transport in the phloem [40]. Similarly, magnesium [41] and calcium [42] levels were critical for high-yield performance, reflecting their respective roles in photosynthesis and cell wall integrity. Notably, variations in the micronutrient manganese (Mn) and the nonessential element barium (Ba) were also observed, suggesting complex, genotype-specific nutrient-uptake profiles beyond the primary macronutrients.

The canonical correspondence analysis (CCA) reinforced that a balanced, multi-nutrient gradient, rather than any single element, primarily drives yield differences among these sugarcane genotypes [43]. K emerged as the primary positive driver of yield, loading strongly on CCA1 and positively associated with taller, thicker, and sweeter stalks [44]. In contrast, elements like boron (B), sodium (Na), manganese (Mn), and silicon (Si) were negatively associated, likely reflecting stress responses or suboptimal nutrient strategies in lower-performing genotypes. While the vast majority of variation was explained by CCA1, a minor secondary axis (CCA2) did separate the genotype GT60, indicating it possesses a unique ionic signature, though its overall impact on yield was limited. The clear ordination of genotypes along the primary axis ultimately illustrates this strong ionomic gradient, suggesting the importance of a well-regulated ionome for achieving optimal growth.

### 3.2. Photosynthetic Pigments and Productivity

The superior biomass production observed in certain sugarcane genotypes appears to be a direct consequence of their enhanced photosynthetic capacity, as evidenced by significantly higher total chlorophyll content. This finding is consistent with previous reports where varieties with higher chlorophyll a, b, and total chlorophyll exhibited the best cane and sugar yields [45,46], underscoring the critical link between photosynthetic efficiency and productivity. Canonical correspondence analysis (CCA) further substantiated this relationship, confirming a coordinated link between nitrogen nutrition, chlorophyll biosynthesis, and yield potential. Given that nitrogen is a central component of chlorophyll, high-yielding genotypes likely possess a more efficient nitrogen-use strategy, which is also reflected in their greener canopies, as indicated by higher SPAD (Soil and Plant Analyzer Development) values, which serve as a proxy for leaf chlorophyll content, and enhanced photosynthetic performance [47]. Notably, GL1215 achieved high yield despite having relatively low chlorophyll levels, implying that factors beyond pigment concentration, such as nutrient-use efficiency or carbon partitioning, may also contribute to its productivity. While less pronounced, genotypic variation in carotenoids. While less pronounced, genotypic variation in carotenoids likely contributes to the overall photosynthetic stability and resilience of these elite varieties through their photoprotective roles. Collectively, these findings reinforce that selecting for traits related to enhanced photosynthetic capacity is a valuable strategy for improving sugarcane productivity, a conclusion further supported by genetic studies identifying QTLs for chlorophyll content as promising selection markers for high-yield breeding [48].

### 3.3. Endogenous Hormone Profiles and Growth Regulation

The genotypic variation in growth traits observed in this study can be attributed to differences in endogenous hormone balances that regulate development and stress responses. Consistent with their established roles, genotypes enriched in growth-promoting hormones such as gibberellins (GAs) and auxins exhibited superior stem elongation and plant height [49]. Variation in auxin-to-cytokinin ratios further contributed to differences in tillering capacity, supporting previous findings that favorable hormonal configurations promoted bud outgrowth and biomass allocation [50]. These results agree with earlier reports suggesting the positive correlation between individual growth hormones and sugarcane vigor; however, our findings extend this knowledge by emphasizing the importance of the relative balance among hormones, rather than the effect of any single hormone alone.

Crucially, our data indicate that yield potential is not dictated by the absolute levels of individual hormones, but rather by the overall hormonal balance and the antagonistic cross-talk between growth- and stress-related pathways. The LT1790 genotype provides a compelling example of this ‘growth-defense trade-off’. Despite having exceptionally high GA_3_ and IAA concentrations that drove rapid stalk elongation, LT1790 simultaneously accumulated elevated abscisic acid (ABA) and salicylic acid (SA). These signals could offset a metabolic shift toward defense readiness and resource conservation, which may have limited yield accumulation. The elevated flavonoid content observed in LT1790 provides further evidence of this metabolic shift toward constitutive defense posture at the expense of sucrose storage [51,52]. The CCA visualization (Figure 6) powerfully corroborates this interpretation. In LT1790, the energetic and signaling investment in its hyperactive defense system—evidenced by its strong association with defense-related vectors in the CCA—is indicative of a metabolic sink that could offset the benefits of its growth promoters, thus limiting its yield realization. This contrasts with some studies in which exogenous GA application consistently boosted both biomass and sugar content [50], underscoring that endogenous hormonal cross-talk can impose stronger internal constraints than single-hormone stimulation suggests.

The CCA reinforced these associations, linking high GA and IAA levels with taller and thicker stalks, while high ABA was negatively associated with growth traits, in agreement with its inhibitory role under stress. Together, these findings suggest that optimizing sugarcane performance requires selecting cultivars with favorable hormonal configurations—such as high GA and cytokinin to drive growth, combined with moderate ABA to ensure constitutive defense posture. This perspective complements previous work showing that genotype-specific responses to hormone treatments significantly influenced cane weight and yield [53], but further demonstrates that breeding for hormonal balance—rather than hormone abundance—could provide a more reliable strategy for improving productivity and adaptability in diverse environments.

### 3.4. Flavonoid Accumulation and Stress Tolerance

The substantial variation in flavonoid content among sugarcane genotypes is consistent with a genotype-dependent’ growth–defense trade-off’, where investment in constitutive defense posture may come at a metabolic cost to productivity. Pigmented varieties, in particular, accumulated markedly higher total flavonoid levels, including anthocyanins, consistent with previous reports that red- or purple-stalked cultivars typically contain far greater anthocyanin concentrations than their green-rinded counterparts. For instance, Hewawansa et al. identified up to 13 distinct anthocyanins in red/purple-stalked varieties, whereas green types showed much lower levels [54].

Beyond pigmentation, flavonoids function as potent antioxidants, scavenging reactive oxygen species (ROS) and protecting photosystems under abiotic stress. The observation that flavonoid-rich genotypes in our study also maintained better chlorophyll retention suggests they are better equipped to handle ambient field stresses. Zhu et al. demonstrated rapid anthocyanin accumulation under chilling stress [55], while drought stress was shown to induce phenolic acid and flavonol accumulation in tolerant sugarcane genotypes. These observations suggest flavonoids as central players in oxidative defense and as markers of a heightened defense readiness [56].

Crucially, the CCA revealed that flavonoid content was not positively associated with higher yield under optimal conditions, quantitatively supporting the trade-off model. This suggests that while enhanced antioxidant capacity confers resilience, it could be associated with reduced sucrose accumulation in unstressed environments. Nevertheless, high-flavonoid genotypes are likely to achieve greater yield stability under adverse conditions, consistent with reports of their superior antioxidant activity, including potent DPPH radical scavenging and DNA protection [57].

Taken together, these findings reinforce that the flavonoid-mediated constitutive defense strategy is a genotype-specific trait and a valuable target for breeding climate-resilient cultivars. Incorporating high-polyphenol germplasm into breeding programs could produce varieties with improved drought, heat, and oxidative stress resistance, thereby stabilizing yields in suboptimal environments. While flavonoid content has not traditionally been prioritized in sugarcane breeding, its dual role in stress protection and prolonged photosynthetic function suggests it will become increasingly important in future improvement strategies.

Beyond analyzing individual components, our data provide a systems-level view of an interconnected regulatory network where ionomic, hormonal, and defensive pathways converge to shape yield potential. The CCA (Figure 6) serves not just as a correlation tool, but as a map of this network, visualizing the dominant synergistic and antagonistic relationships.

### 3.5. Implications for Breeding High-Yield, High-Resilience Cultivars

The case of LT1790 is best understood not simply as a trade-off, but as a clear example of network-level antagonism. In this genotype’s integrated network, the strong growth-promoting signals from the ‘hormone node’ (high GA_3_ and IAA) appeared to be counterbalanced by a hyperactive ‘defense node’ (high ABA, SA, and flavonoids). This suggests a crucial principle: the network’s final output (yield) is not determined by the strength of a single input, but by the net balance of competing signals across different functional modules. Even the strong, positive signal from the ‘ionome node’—specifically the high potassium (K) levels in LT1790 that should synergize with growth—was insufficient to overcome the powerful inhibitory effect of the defense node. This interaction is consistent with the idea that yield is an emergent property of the entire network, where antagonistic cross-talk can override the benefits of otherwise favorable individual traits.

Collectively, our findings suggest that sugarcane breeding requires a shift from conventional selection for yield components alone toward a more integrated approach that incorporates underlying physiological and metabolic traits. Elite performance is not achieved by maximizing individual pathways, but by striking a balance between growth, defense, and nutrient utilization. For example, selecting for nutrient-use efficiency (NUE) could be particularly valuable: varieties that maintain high yield despite lower tissue nitrogen concentrations would be advantageous in nutrient-limited soils, whereas in high-fertility systems, cultivars capable of rapidly assimilating nutrients without compromising structural integrity would maximize returns [58]. Identifying genotypes that combine efficient uptake with robust lodging resistance could guide more targeted breeding strategies.

Hormonal balance emerged as another critical determinant of yield potential. While high gibberellin (GA) and auxin (IAA) levels promoted rapid stem elongation, our results with LT1790 suggest that simply maximizing growth hormones is not sufficient. In this genotype, co-accumulation of stress hormones (ABA, SA) antagonized growth signals, redirecting resources toward constitutive defense and limiting sucrose storage. This underscores the need for breeding goals to evolve from enhancing hormone abundance to optimizing hormonal balance. Integrating hormone-related markers—such as DELLA proteins linked to GA sensitivity—into selection indices could help identify cultivars that achieve vigorous growth without costly defense activation [59].

## 4. Materials and Methods

### 4.1. Plant Materials

Six commercial sugarcane (*Saccharum* spp.) genotypes with contrasting yield and sucrose content were used in this study, including ZZ14, GL1215, GL05136, LT1790, GT59, and GT60. These genotypes were obtained from the Guangxi Academy of Agricultural Sciences (Nanning, Guangxi, China) and are widely cultivated in southern China. Preliminary field surveys indicated that ZZ14 and GL1215 are high-yielding and high-sucrose cultivars, whereas GT59 and GT60 show lower yield potential, and LT1790 and GL05136 display intermediate performance. All plants were maintained under standard greenhouse conditions prior to experimental treatments.

### 4.2. Sample Collection and Preparation

Leaf samples were collected from the main stems of the six genotypes during the peak elongation stage (approximately 150–180 days after planting) in a controlled nursery environment. For each biological replicate (*n* = 3 per genotype), fresh leaves were longitudinally bisected along the midrib to create two morphologically matched aliquots. The first aliquot was immediately snap-frozen in liquid nitrogen (LN_2_) and stored at −80 °C for subsequent phytohormone and flavonoid content analyses. The second aliquot was oven-fixed at 105 °C for 2 h to deactivate enzymes and then dried at 65 °C (±1 °C) until constant weight (mass variation < 0.2% over 24 h) for subsequent physicochemical characterization and ionomic (elemental) analysis by ICP-MS.

### 4.3. Agronomic Parameter Quantification

At physiological maturity (310–330 days post-planting), four key yield-related traits were measured: height, stem diameter, stalk biomass production (kg/ha), and soluble solids content (Brix, °) of expressed juice. For each genotype, three biological replicates (*n* = 3) were used, and within each replicate, measurements were taken from three representative stalks (total *n* = 9 per genotype). Brix levels were determined using a temperature-compensated digital refractometer (Atago PAL-1, 0–53% Brix range, calibrated at 20 °C), providing a direct measure of sucrose concentration.

### 4.4. Ionomics Analysis

Dried and finely ground leaf tissue samples were used for ionomic profiling. Ionomics analysis was conducted following the method of Ilieva et al. [60]. Dried samples were homogenized to a fine powder using a stainless-steel grinder, then sieved through a 100-mesh (150 μm) nylon sieve to ensure uniform particle size. For digestion, 0.3 g (±0.1 mg) of dried powder was weighed into quartz vessels and treated with 8 mL of concentrated nitric acid (65%). Digestion was performed in a super microwave digestion system (SUPEC EXPEC 790S, Expec-Tech, Hangzhou, China) using the following protocol: ramping from 25 °C to 240 °C over 20 min, holding at 240 °C for 30 min, and cooling to 50 °C. The resulting digests were filtered through 0.45 μm PTFE membranes, diluted to 50 mL with ultrapure water, and analyzed via inductively coupled plasma optical emission spectrometry (ICP-OES; EXPEC 6500, Expec-Tech, Hangzhou, China) for direct counts per second (CPS) measurement.

### 4.5. Plant Hormone Analysis

The snap-frozen leaf samples were used for phytohormone analysis because they are metabolically active and better reflect physiological differences among genotypes. Phytohormone extraction and quantification were conducted using a UHPLC-MS/MS system (Thermo Fisher, Waltham, MA, USA), based on the method of Kojima et al. [61] with slight modification. Approximately 100 mg of finely ground sugarcane tissue was extracted with a pre-cooled methanol: formic acid solution (80:19:1, v/v/v) containing internal standards for each target hormone to ensure accurate quantification. Samples were vortexed, sonicated for 30 min at 4 °C, and centrifuged at 12,000× *g* for 15 min to remove insoluble debris. The supernatant was collected, and the residue was re-extracted twice to maximize recovery. The combined extracts were dried under nitrogen gas and reconstituted in 200 µL of 50% methanol before UHPLC-MS/MS analysis. Chromatographic separation was achieved on a reversed-phase column with a gradient elution program optimized for the target compounds. Hormones were detected in multiple reaction monitoring (MRM) mode, with ion pairs, collision energies, and mass spectrometry settings optimized as specified in Appendix A.

### 4.6. Flavonoid Analysis

The same snap-frozen leaf samples were used for flavonoid analysis because they are metabolically active and can accurately represent the in vivo accumulation status of secondary metabolites. Flavonoid extraction and quantification were performed using a UHPLC-MS/MS system (Thermo Fisher, USA), following the modified protocol of Wang et al. [62] with minor adjustments. Approximately 100 mg of finely ground sugarcane tissue was extracted with 70% methanol containing an internal standard to ensure accurate quantification. Samples were sonicated for 30 min at 4 °C to enhance extraction efficiency, then centrifuged at 12,000× *g* for 10 min to remove insoluble debris. The resulting supernatant was filtered through a 0.22 µm membrane before UHPLC-MS/MS analysis. Chromatographic separation was achieved using a reversed-phase column under gradient elution conditions optimized to resolve a wide range of flavonoid compounds. Flavonoids were detected in multiple reaction monitoring (MRM) mode, with ion pairs, collision energies, and mass spectrometry settings specified in Appendix A.

### 4.7. Photosynthetic Pigments Analysis

Chlorophyll a, chlorophyll b, and total carotenoid contents were quantified using a modified Zeb [63] procedure. About 100 mg of frozen tissue was ground under liquid nitrogen and first extracted with 5 mL of ice-cold acetone by orbital shaking (130 rpm, 60 min). After centrifugation (10,000× *g*, 10 min, 4 °C), the supernatant was collected and the pellet re-extracted with 10 mL of absolute ethanol by vortexing for 30 min, followed by another centrifugation step. Extractions were repeated until the residue was colorless, and all supernatants were pooled. Solvents were removed under vacuum at 35 °C, the dry residue re-dissolved in 2 mL of methanol, and the solution clarified by filtration before injection. Analyses were carried out on an HPLC system with PDA detection using a C18 column at 25 °C, eluted with a methanol–water–methyl tert-butyl ether gradient. Detection was set at 450 nm for carotenoids and 650 nm for chlorophylls, and peaks were identified by matching retention times and spectra to authentic standards.

### 4.8. Statistics and Data Analysis

All statistical analyses were performed using SPSS (v. 26, IBM, USA), SIMCA (v. 14.1, Sartorius, Sweden), and R (v. 4.2.1, R Foundation). A *p*-value < 0.05 was considered statistically significant for all tests.

For univariate analysis, a one-way analysis of variance (ANOVA) was used to test for significant differences in agronomic traits, pigment content, hormone levels, and flavonoid concentrations among the six sugarcane genotypes. When the ANOVA indicated a significant effect (*p* < 0.05), a post hoc Duncan’s multiple range test was performed to identify which specific genotype means were significantly different from one another.

To explore the complex relationships within the high-dimensional ionomic, hormonal, and flavonoid datasets, several multivariate statistical techniques were employed. Initially, Principal Component Analysis (PCA) was used for an unsupervised exploration of the ionomic data to visualize overall variation and natural clustering patterns. Prior to PCA, the data were Pareto (Par) scaled to reduce the influence of high-abundance elements while retaining important variation. To identify the specific variables responsible for discriminating between pairs of genotypes, supervised Orthogonal Partial Least Squares Discriminant Analysis (OPLS-DA) was performed. The validity of each OPLS-DA model was assessed using the cumulative Q^2^ (Q^2^cum) value from seven-fold cross-validation, with models having a Q^2^cum ≥ 0.5 considered robust. Variables with a high Variable Importance in Projection (VIP) score (VIP > 1.0) were identified as the key discriminators.

Finally, to investigate the integrated relationships between biochemical profiles and agronomic performance, Canonical Correlation Analysis (CCA) was employed using the CCA package in R. Two separate CCA models were constructed: one linking the ionomic data to yield traits (plant height, stem diameter, yield, and sugar content), and another linking the combined phytohormone and flavonoid data to the same yield traits.

## 5. Conclusions

This study demonstrates that genotypic differences in nutrient use efficiency, hormonal regulation, and flavonoid-mediated defense are closely linked to sugarcane yield potential and reflect distinct, constitutive strategies for defense investment. Unlike previous studies that often focused on single traits, the integrated CCA approach reveals that high-yielding sugarcane genotypes typically achieve a superior metabolic balance. This includes efficient nutrient uptake, characterized by higher accumulation of potassium (K), magnesium (Mg), and calcium (Ca); a favorable hormone profile that balances growth-promoters like gibberellins (GAs) against stress-related hormones like abscisic acid (ABA) to avoid costly growth-defense trade-offs; abundant photosynthetic pigments such as chlorophyll a and b; and a robust antioxidant system that avoids the excessive accumulation of defense-related flavonoids like Quercetin and rutin. This holistic perspective underscores the importance of selecting for the optimal balance among these physiological traits in breeding programs, offering a roadmap for developing next-generation cultivars with high productivity and enhanced stress tolerance.

## Figures and Tables

**Figure 1 plants-14-03181-f001:**
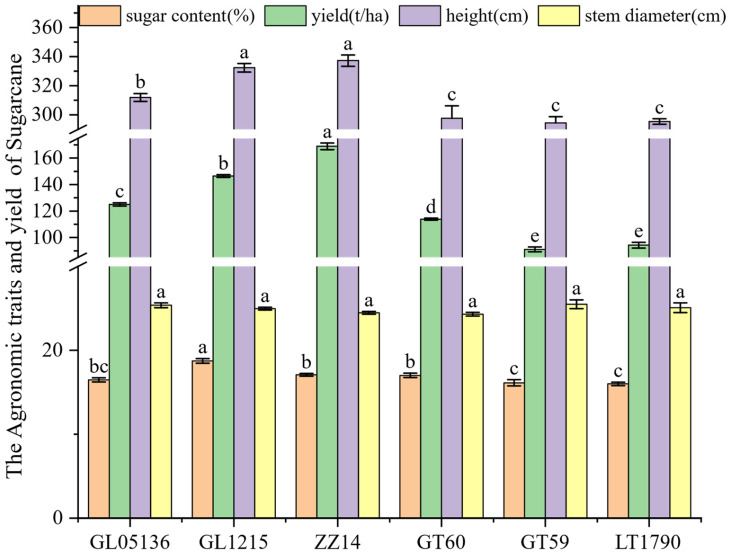
Plant height, yield traits, sucrose content and juice Brix of six sugarcane cultivars. Note: Bar heights indicate the mean values of 3 biological repetitions. Error bars represent the standard error of the mean (SE). Different lowercase letters above the bars indicate significant differences among varieties (ANOVA, *p* < 0.05).

**Figure 2 plants-14-03181-f002:**
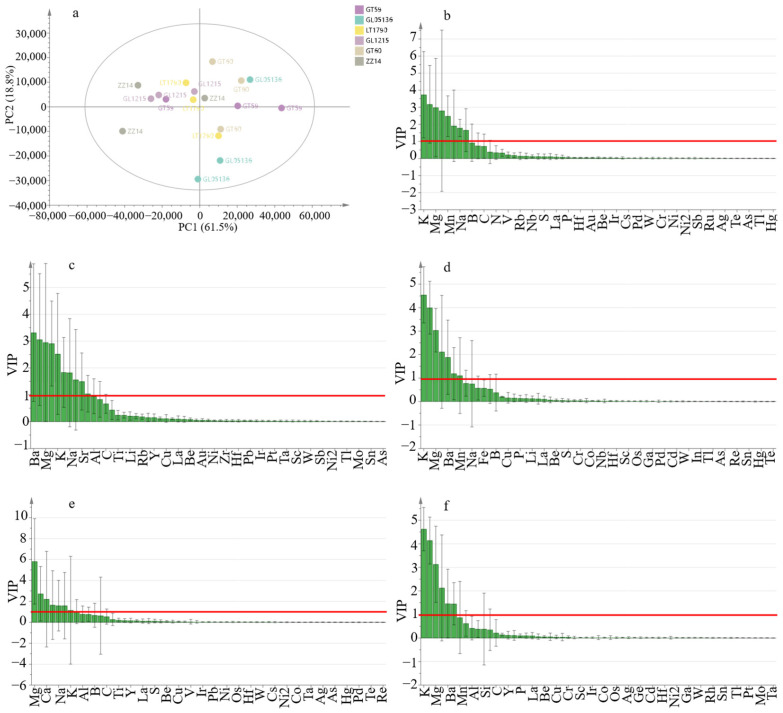
Multivariate ionomic analysis of six sugarcane genotypes. (**a**) Principal component analysis (PCA) of ionomic profiles showing the clustering patterns of genotypes. (**b**) OPLS-DA: GT59 vs. LT1790. (**c**) OPLS-DA: GL05136 vs. GL1215. (**d**) OPLS-DA: GL05136 vs. GT60. (**e**) OPLS-DA: GL05136 vs. ZZ14. (**f**) OPLS-DA: GT59 vs. GT60. Key discriminating elements in each model are indicated by variable importance in projection (VIP) scores, and each dot represents one biological replicate (*n* = 3 per genotype).

**Figure 3 plants-14-03181-f003:**
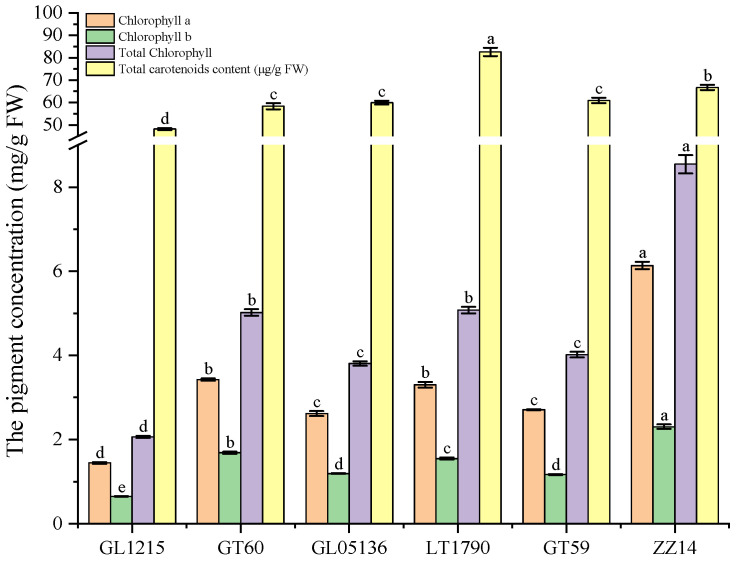
Genotype-dependent variation in photosynthetic pigment content across sugarcane varieties. Note: Carotenoid values represent total carotenoid content rather than individual compounds. Different lowercase letters above the bars indicate significant differences among varieties (ANOVA, *p* < 0.05)

**Figure 4 plants-14-03181-f004:**
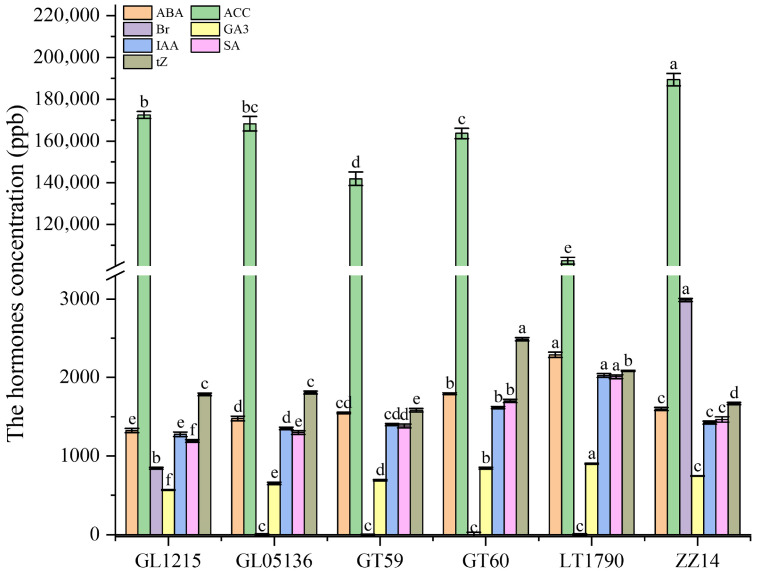
Endogenous phytohormone content in six sugarcane genotypes. Different lowercase letters above the bars indicate significant differences among varieties (ANOVA, *p* < 0.05)

**Figure 5 plants-14-03181-f005:**
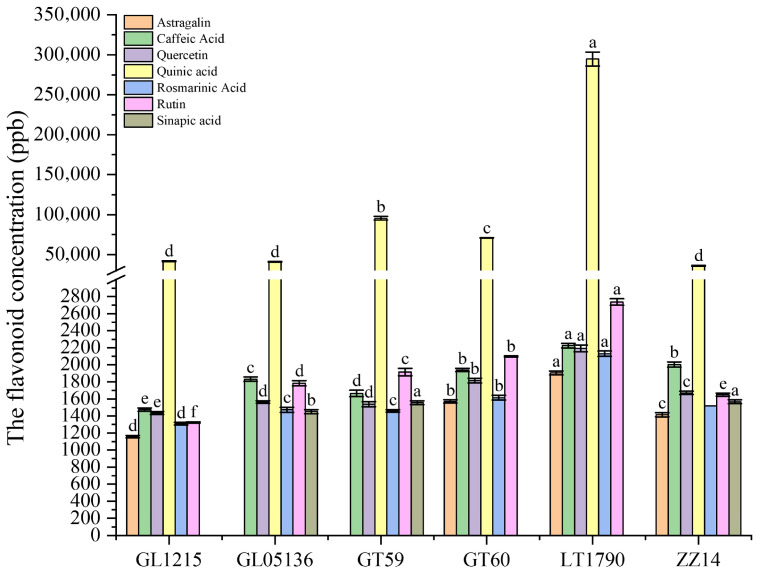
Genotype-dependent variation in leaf flavonoid composition among sugarcane varieties. Different lowercase letters above the bars indicate significant differences among varieties (ANOVA, *p* < 0.05)

**Figure 6 plants-14-03181-f006:**
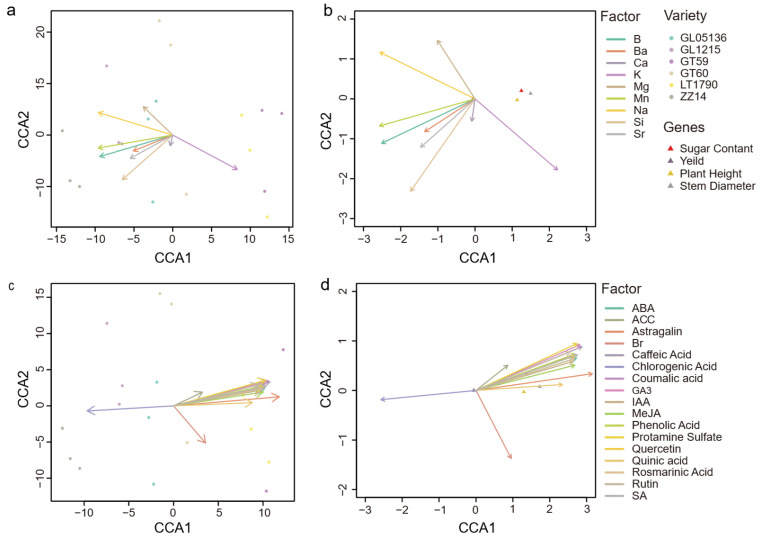
Canonical correlation analysis (CCA) linking yield-related traits with (**a**,**b**) ionomic and (**c**,**d**) metabolite data in sugarcane. Separate CCA models were constructed using (**a**,**b**) mineral element (ionomic) variables and (**c**,**d**) phytohormone/metabolite variables to examine their relationships with yield components. Panels (**a**,**c**) show biplots including genotype samples, where each dot represents one biological replicate (*n* = 3 per genotype). Panels (**b**,**d**) display biplots of variable loadings and yield components (arrows). Vectors indicate the direction and strength of associations with the canonical axes.

## Data Availability

All data generated or analyzed during this study will be available upon reasonable request.

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
