# Peer review of "The Ionome–Hormone–Flavonoid Network Shapes Genotype-Dependent Yield Adaptation in Sugarcane"

_plants, 2025, doi:10.3390/plants14203181_

Round 1
Reviewer 1 Report
Comments and Suggestions for Authors The work is very interesting and represents a significant advance in order to modify sugarcane yields. However, many aspects must be considered to improve the quality of the ms. Regarding abstract, it did not summarize the most important results and did not clearly reflect the main findings. Introduction must be rewrited, there are repeated paragraphs, besides it is too long, several references are not neccesary according to the topic of the ms. Even, references must be reorder to follow numeration. In addition, many wrong or unexplained information is mentioned. As several acronyms are included, it is a litle bit difficult to read it, they should be described the first time they appeared. Even the structure of the section should be modify to follow a logical order that leads readers to understand the gap in the knowledge. Respect to methodology section, the assyas need to be explained whith more details, besides there are some mixed information. Please revise the section and describe assays according they were carried out, especially for yield trials. There are some conceptual errors, since commercial hybrids are interespecifc ones, they do not belong to officinarum genus. Result presentations must be greatly improved. Figures must be modified to better present results, i.e. references are different from colors in figure, besides the text descibed the same information that figures. The way in which results are shown must be greatly improved, i.e. in PCA, each genotype is represented by several dots, why? Discussion has many repeated results, besides they should be discussed according the order in which they were obtained. Another aspects that should be considered is that authors mentioned in several ocassions stress tolerance; however, they do not perform any assay about it, so they do not associate results to stress toletance. In addition, it is suggested to write the ms using 3rd person instead of the first one.

I think that english language is ok, it is not the main aspect that must be improved.
Author Response
Dear Reviewer,
We sincerely thank you for your constructive and valuable comments on our manuscript.
Please kindly find our detailed point-to-point responses in the attached file entitled “Response to Reviewer 1”.
We hope our revisions and explanations satisfactorily address all your comments.
Thank you again for your time and thoughtful review.

Reviewer 2 Report
Comments and Suggestions for Authors
The results of the presented research seem quite interesting; however, the article cannot be accepted in its present form. It must be rewritten in order to make the results clearer for the reader.
Several problems which I observed:
- I cannot see how the statements of the abstract given in lines 23-26 are justified by the results; for example, if we compare the figures 1 and 4, we can see that, in contrast to what the authors claim, the genotype with the highest gibberellin and auxin concentrations (LT1790) had lower yield than most of the other investigated genotypes.
- The text in lines 84-91 is identical to that in lines 92-99. However, different references are cited at the identical statements (references no. 22-23 in the first set of text, and references no. 24-25 in the repetition of the same text).
- In line 167, it is stated that "we compared four sugarcane cultivars", although the results are given from six cultivars.
- Figure 1 is very problematic. Different parameters with different units of measurement are lined along the the same scale. This makes some of them almost disappear inside such a graph. Besides, it remains unclear what the column bars in light blue and pink colours signify.
- The elements inside Figure 2 are hardly visible, at least in the pdf variant.
- Considering Figure 3 and the statement in line 603 (in sub-chapter 4.6.), it remains unclear whether the total carotenoid content was analysed, or only lutein.
- The text in lines 621-628 seems to be retained from a template and not as coming from this particular research.
- The statement in lines 633-635 seems too general. It would be more helpful if some of the particular chemicals were highlighted here.
Author Response
Dear Reviewer,
We sincerely appreciate your insightful and helpful comments on our manuscript.
Please kindly find our detailed point-to-point responses in the attached file entitled “Response to Reviewer 2”.
We hope our revisions and explanations have fully addressed your concerns.
Thank you again for your time and effort in reviewing our work.

Round 2
Reviewer 1 Report
Comments and Suggestions for Authors
The ms has been greatly improved in its new version. Authors took into account all suggestions, properly responding comments. Only minor details were included in the attached file.

Author Response
Dear Reviewers, Thank you for your helpful comments. We have revised the manuscript according to all suggestions. A detailed point-by-point response is uploaded as a separate file. All changes in the manuscript are highlighted in red for easy identification.
